# Histology of the Upper Gastrointestinal Tract, Morphometry and Lymphocyte Subpopulations of the Duodenal Mucosa: Insights from Healthy Individuals

**DOI:** 10.3390/ijms26031349

**Published:** 2025-02-05

**Authors:** Albert Martín-Cardona, Anna Carrasco, Carme Ferrer, Clarisa González-Mínguez, Luis Luizaga-Velasco, Xavier Tarroch, Gerardo Gonzalez-Puglia, Eva Tristán, Natalia Berenice Cardozo-Rembado, Natàlia Pallarès, Cristian Tebé, Beatriz Arau, Isabel Salvador, Ingrid Fajardo, Raimon Rifà, Laura Ruiz, Pablo Ruiz-Ramírez, Sònia Fernández-Herrera, Agnès Raga, Montserrat Aceituno, Yamile Zabana, Carme Loras, Mireia Fonolleda, Jordi Roigé, Fernando Fernández-Bañares, Maria Esteve

**Affiliations:** 1Digestive Diseases Department, Hospital Universitari Mútua Terrassa, University of Barcelona, 08221 Terrassa, Catalonia, Spain; anna.carrasco.garcia@gmail.com (A.C.); gerardgonzalezpuglia@gmail.com (G.G.-P.); etrislo@hotmail.com (E.T.); ncardozo@mutuaterrassa.es (N.B.C.-R.); beatrizarau@mutuaterrassa.es (B.A.); isalvador@mutuaterrassa.cat (I.S.); ifajardo@mutuaterrassa.cat (I.F.); rrifa@mutuaterrassa.cat (R.R.); lruiz@mutuaterrassa.es (L.R.); pruiz@mutuaterrassa.es (P.R.-R.); araga@mutuaterrassa.cat (A.R.); maceituno@mutuaterrassa.es (M.A.); yzabana@mutuaterrassa.es (Y.Z.); cloras@mutuaterrassa.cat (C.L.); ffbanares@mutuaterrassa.es (F.F.-B.); 2Centro de Investigación Biomédica en Red de Enfermedades Hepáticas y Digestivas (CIBERehd), Instituto de Salud Carlos III, 28029 Madrid, Spain; 3Pathology Department, Hospital Universitari Mútua Terrassa, University of Barcelona, 08221 Terrassa, Catalonia, Spain; carmeferrer@mutuaterrassa.es (C.F.); cgonzalez@mutuaterrassa.es (C.G.-M.); lluizaga@mutuaterrassa.cat (L.L.-V.); xtarroch@mutuaterrassa.cat (X.T.); 4Biostatistics Support and Research Unit, Germans Trias I Pujol Research Institute and Hospital (IGTP), 08916 Badalona, Catalonia, Spain; npallares@igtp.cat (N.P.); ctebe@igtp.cat (C.T.); 5Department of Immunology, Catlab, 08232 Viladecavalls, Catalonia, Spain; mfonolleda@catlab.cat; 6Department of Genetics, Catlab, 08232 Viladecavalls, Catalonia, Spain; jroige@catlab.cat

**Keywords:** healthy mucosa, eosinophils, mast cells, intestinal morphometry, intraepithelial lymphocytes, γδ+ cells, natural killer cells, innate lymphoid cells, flow cytometry

## Abstract

The upper oesophagogastrointestinal (UEGI) tract histology, intestinal morphometry and lymphocyte subpopulations of healthy people is scarcely known. In research studies of inflammation involving the UEGI tract, there is a lack of adequate healthy controls. Aims: To evaluate the histology of the UEGI tract and the duodenal lymphocyte subpopulations of healthy volunteers and patients with gastroesophageal reflux disease (GERD), the latter to assess if it could replace healthy subjects. Healthy individuals were excluded if they had symptoms, comorbidities, pregnancy, toxics, medications or abnormal blood analysis. Subjects in both groups with abnormal duodenal intraepithelial lymphocyte (IEL) counts were also excluded. A total of 280 subjects were assessed, and 37 were included (23 healthy and 14 with GERD). The GERD group showed a higher IEL count (median [IQR]: 19.5 [17–22]), than healthy group: (15 [12–18]), *p* = 0.004. Eosinophils, mast cells and intestinal morphometry were similar in both groups. In the lamina propria, CD4+ T cells decreased (*p* = 0.008), and CD8+ T cells increased (*p* = 0.014). The total innate lymphoid cells (ILC) and CD3− cells decreased (*p* = 0.007) in GERD group compared to healthy controls. At the intraepithelial level, NKT cells increased (*p* = 0.036) and ILC3 decreased (*p* = 0.049) in the GERD group. This is the first study to comprehensively map the histology, morphometry and duodenal subpopulations of healthy volunteers to help define a “gold standard” of normality. The differences found between both groups suggest that, whenever possible, healthy subjects should be included in research studies. Alternatively, we can consider a well-defined homogenous group with GERD to serve as the control group.

## 1. Introduction

Clinical and pathological findings identify different diseases. Systematic histological studies of the digestive tract provide a wealth of information; however, in many cases, these pathological findings are nonpathognomonic, and the clinical presentations of different diseases (dyspepsia, anaemia, diarrhoea, etc.) can be indistinguishable. Therefore, additional techniques based on disease-specific pathophysiological findings are necessary for more precise diagnosis.

A good example of a disease-specific immunological marker is the increased percentage of TCRγδ+ cells found in the intraepithelial mucosa of the duodenum of coeliac patients (named the coeliac lymphogram pattern) [1]. This pattern is a very useful tool for diagnosing seronegative coeliac disease or lymphocytic coeliac enteropathy (generally seronegative); additionally, it allows a differential diagnosis with other diseases, such as Crohn’s disease and *Helicobacter pylori* infection, among others, that may have overlapping symptoms and identical histopathological features at the duodenum. Therefore, lymphocyte subpopulation patterns may guide the aetiology of diseases based on their pathophysiology.

When defining pathologically associated disease patterns, one important issue is the lack of a gold standard of what is considered healthy controls. Most studies evaluating the lower digestive tract mucosa include “healthy control” subjects undergoing colonoscopy for colorectal cancer screening if the colonic mucosa is macroscopically and microscopically normal [2]. In Western countries, there are no mass screening programmes for oesophagogastric cancer due to its relatively low prevalence. Thus, routine endoscopic assessment of the upper gastrointestinal tract with biopsies in patients without digestive symptoms in our environment is lacking. In fact, the limit of normal lymphocyte infiltration in the duodenum for diagnosis of coeliac disease (CD) [3,4,5,6] or the normal value of eosinophils in the oesophagus for oeosinophilic oesophagitis (EoE) [7] was not established until recently. However, even in these cases, the included controls considered “normal” had digestive symptoms [4]. Information regarding the patterns of lymphocyte subpopulations in the intestinal tract of healthy patients is scarce [8]. Additionally, studies included patients with functional digestive disorders, such as irritable bowel syndrome (IBS), as the control group [8]. IBS is characterised by bloating, abdominal pain, and alterations in bowel habits, which can range from constipation to diarrhoea or both [9]. Although IBS is generally characterised by normal intestinal histology, many studies have demonstrated some type of low-grade mucosal inflammation [10], with mast cells playing a role in its pathophysiology [11,12,13,14]. In addition, lymphocyte subpopulation patterns remain unknown [15].

Having a good standard of normality is essential for research studies. This standard should be established in individuals without clinical symptoms or the presence of any disease. Therefore, the aim of this study was to assess the histology (lymphocytes, eosinophils, and mast cells) of the upper digestive tract (oesophagus, stomach, and duodenum), morphometry and lymphocyte subpopulations of the duodenal mucosa in asymptomatic healthy individuals. As a secondary aim, we assessed patients with gastroesophageal reflux disease (GERD) to ascertain whether they had a healthy duodenum that allowed them to serve as a control group for diseases involving this part of the intestinal tract.

## 2. Results

### 2.1. Baseline Characteristics of the Study Population

A total of 280 subjects were assessed, and 37 were included: 23 healthy individuals (56.5% female, 24.7 ± 4.2 years) and 14 patients with GERD (57.1% female, 33.3 ± 14.1 years) (Figure 1. Study flowchart). No differences in sex were found, but healthy controls were significantly younger than those in the GERD group (*p* = 0.022).

The GERD group included a slightly greater percentage of smokers than in the healthy control group, and PPIs was used by more than three-quarters of them. Neither patients with GERD nor healthy controls took any other medication, nor did they consume alcohol. All GERD patients and healthy controls had negative CD serology. In the GERD group, five patients (50%) were DQ2.5+, whereas the healthy controls had low-risk or negative CD genetics. In terms of endoscopic findings, 21% of the GERD patients had reflux oesophagitis. No adverse events occurred during upper gastrointestinal endoscopy. Table 1 describes the baseline characteristics of the healthy individuals group and the GERD group.

### 2.2. Histological Features (Lymphocytes, Eosinophils and Mast Cells)

Table 2 describes the histological characteristics of the oesophageal, gastric and duodenal mucosa in the healthy and GERD groups. Two cases of eosinophilic oesophagitis were observed in the GERD group, and a similar percentage of mild chronic gastritis was noted in both groups (*p* = 0.200). No differences were found in eosinophil or mast cell counts in the oesophageal, gastric or duodenal mucosa (*p* = ns). For the IEL count, no differences were found in the oesophageal and gastric mucosa; however, a significantly greater percentage of duodenal IELs (median [IQR]: 19.5 [17–22]) were found in patients with GERD than in healthy controls (median [IQR]: 15 [12–18]), *p* = 0.005. However, both groups had normal IEL counts in the duodenum, as established by the inclusion criteria. No parasites were identified on the duodenal surface of healthy controls or patients with GERD.

In Table 3, the histological characteristics of healthy individuals analysed by sex are described. No significant differences were observed in the histological characteristics studied between men and women in healthy individuals, except for the duodenal mast cell count, which was increased in women (*p* = 0.009). Additionally, no significant differences were found in the histological characteristics when the data were analysed by age separated into two groups by the median value (<25 versus ≥25 years) in healthy individuals.

Figure 2 shows histological images of duodenal lymphocyte, eosinophil, and mast cell counts of healthy individuals.

### 2.3. Duodenal Morphometry

Despite a greater IEL count in the duodenal mucosa in patients with GERD, no differences were observed between the healthy and GERD groups in terms of villus morphometry (villus height [μm] and crypt depth [μm]) (Table 4 and Figure 3). The architecture and shape of the villi (villus height-to-crypt depth ratio) were preserved in both groups. Figure 4 shows histological images of the duodenal mucosa with measurements of villus height and crypt depth to assess duodenal morphometry.

### 2.4. Intestinal Lymphocyte Subpopulations

Table 5 shows the intestinal lymphocyte subpopulations in healthy individuals compared with those in the GERD group, separated by the two intestinal compartments, the intraepithelial compartment and lamina propria.

The most remarkable differences between the healthy control group and patients with GERD were found in the lamina propria compartment, with a significant decrease in CD4+ T cells (*p* = 0.008) and an increase in CD8+ T cells in patients with GERD compared with those in healthy controls (*p* = 0.014). The total number of innate lymphoid cells (ILC) also decreased in the lamina propria of patients with GERD (*p* = 0.007), mainly due to a reduction in the number of ILC1 (*p* = 0.012). At the intraepithelial level, ILC3 significantly decreased (*p* = 0.049) and natural killer T (NKT) cells increased (*p* = 0.036) in the GERD group compared to healthy controls.

Two of the subpopulations evaluated were those that conform to the very specific immunological signature related to CD at the intraepithelial level, which has been named the coeliac lymphogram. No differences in the number of TCRγδ+ and CD45+CD3− cells were detected between the healthy and GERD groups. However, differences were noted in CD45+CD3− in the lamina propria, with a decrease in the GERD group (*p* = 0.007).

Figure 5 shows the distribution of lymphocyte subpopulations (%), for which there was a significant difference in the lamina propria or intraepithelial compartment between groups. Additionally, the two subpopulations of the coeliac lymphogram are represented. The more overlapping curves there are, the fewer differences there are between the healthy and GERD groups. Figure 6 shows intestinal cytometry panels of major intestinal lymphocyte subpopulations in healthy individuals compared to the GERD group.

## 3. Discussion

This is the first study assessing the morphological findings of the upper gastrointestinal tract and lymphocyte subpopulations by flow cytometry in the duodenal mucosa of strictly asymptomatic healthy individuals. Furthermore, duodenal morphometry was used for a more accurate evaluation [16,17]. The reason for the lack of studies of normal healthy intestines is the difficulty in finding asymptomatic true healthy controls, especially regarding the upper gastrointestinal tract. In fact, after exhaustive evaluation of more than 100 potential healthy controls, we could include only one in every six subjects. Even in this case, four patients had mild antritis. These patients were ultimately included because the duodenal mucosa was macroscopically normal and the number of IELs was below the considered optimal cut-off point of 25 IELs/100 enterocytes.

Patients with pure GERD symptoms could theoretically be good controls, especially for studies assessing the duodenal mucosa. There were no significant differences in mast cell, eosinophil or lymphocyte counts between healthy controls and patients with GERD in the oesophagus and stomach. At the duodenal level, no differences were found in eosinophil or mast cell counts or in duodenal morphometry. The latest ESPGHAN/NASPGHAN guidelines for eosinophilic gastrointestinal disorders suggest normal eosinophil thresholds of 15, 30 and 50 for the oesophagus, stomach and duodenum, respectively [7,18,19,20]. Nevertheless, in the group of healthy individuals from the present study, the mean eosinophil counts were much lower, as they were strictly asymptomatic. Mast cells may play a significant role in the pathophysiology of IBS [11,12,15]. Studies in this field should consider the mast cell count found in our study as a reference of normality.

Incorporating a sex and gender perspective in biomedical research, the histological data were disaggregated and analysed by sex. A significant increase in the duodenal mast cell count was found in women (*p* = 0.009). This finding is consistent with previous literature, such as the studies by Barbara et al. [21] and Cremon et al. [22], which also identified a significant increase in the number of mast cells in the colonic mucosa, predominantly in women with IBS. Physiologically, mast cells express receptors for oestrogen and progesterone, suggesting that these hormonal mediators could modulate their activity [21,22]. This phenomenon may be related to hormonal fluctuations during the menstrual cycle in women, contributing to a higher mast cell density in the female intestinal mucosa. No significant differences were found in the remaining histological characteristics between males and females.

With respect to duodenal morphometry, no differences were observed between the healthy individuals and the GERD group. We cannot determine whether an increase in sample size would render this difference significant; however, the results obtained are consistent with those previously published by Rostami et al., where no differences in the villus height-to-crypt depth ratio were observed between the two studies [17]. We recommend that studies assessing mucosal structure in the duodenum provide data on villous morphometry.

However, differences were observed in IEL counts, which were greater in the GERD group than in the healthy control group. This finding could be explained by increased acid reflux at the duodenal level or by the use of PPIs. Acid reflux in the duodenum can cause histological lesions in the duodenal bulb and the second portion of the duodenum [23]. Additionally, chronic PPI use may lead to alterations in the gastric microbiota [24] and bacterial overgrowth [25], both of which could result in increased IEL counts in the duodenum [23,26].

The “normal” cut-off for IELs in the duodenum has been a topic of debate in the medical literature [27]. The most important limitation is the lack of a universal definition of what is considered ‘normal’. The interest in having a cut-off was focused mainly on the diagnosis of mild enteropathy due to CD. The first stage of this enteropathy, triggered by gluten ingestion, is characterised by diffuse infiltration of intestinal villi by IELs. Although some other diseases may produce similar histopathological lesions, the magnitude of the increase seemed to be greater in CD than in other pathological conditions. For this reason, and because of the lack of true healthy controls, all studies assessing the “normal” cut-off of IEL were performed by comparing patients with CD with a variety of disease controls. These patients included those with dyspepsia, bloating, diarrhoea, GERD and others. Using these controls, the most agreed-upon cut-off was established with 25 IELs/100 epithelial cells. In our study, we found a median value of 15 IELs/100 epithelial cells in healthy subjects, with a maximum value of 20 IELs/100 epithelial cells. This normal healthy pattern should be considered in studies assessing the immune cell response in diseases other than CD.

Regarding the duodenal subpopulations evaluated, the most important differences between healthy controls and patients with GERD were found in the lamina propria compartment; this consisted of profound differences in the CD4/CD8 balance, with a significant decrease in CD4+ T cells and an increase in CD8+ T cells in patients with GERD compared with those in controls. In addition, there was a decrease in CD45+CD3− and ILC (primarily due to ILC1) subpopulations in patients with GERD. In the intraepithelial compartment, ILC3 had a differential profile in GERD patients compared with that in healthy controls, with a significant reduction in patients with GERD. Moreover, GERD is associated with an increase in cytotoxic NK cells, mainly due to a significant increase in NKT (CD3+CD56+) cells. All these findings, particularly the increase in lamina propria CD8+ T cells and intraepithelial NKT cells, demonstrates a predominant activation of innate immune response in the duodenum of patients with GERD compared to healthy subjects [28]. These differences are likely explained by acid reflux and/or PPI use [24,25]. By contrast, we lack a clear explanation for the reduction in total ILCs and ILC1 in the lamina propria, as well as ILC3 in the intraepithelial compartment, in patients with GERD compared to healthy controls. Certain bacterial overgrowth induced by PPIs or acid-induced inflammation would be expected to have the opposite effect [29,30,31]. In healthy subjects, the relative composition of ILCs aligns with previous descriptions, showing a predominance of ILC1 in the intraepithelial layer of the duodenum and ILC3 in the lamina propria, whereas ILC2 was undetected in both healthy controls and patients with GERD. In fact, under homeostatic conditions, ILC2 primarily resides in adipose tissue, the lungs, and the skin rather than the intestine [29].

Increased intraepithelial TCRγδ+ and decreased NK CD3− cells, known as a coeliac lymphogram, is a characteristic hallmark of the immunological response of CD patients and is very useful for the diagnosis of challenging cases [1]. We did not find differences between healthy controls and patients with GERD regarding these subpopulations. These findings suggest that patients with GERD could be acceptable controls for studies involving the immune response of CD, particularly in the epithelial compartment.

The selection of the two groups assessed in the present study was based on CONSORT [32], STARD [33] and QUADAS-2 [34] recommendations that establish the need for clear inclusion and exclusion criteria defining control populations. In this sense, the groups we included in the present study are homogeneous, representative of the target population, comparable, and selected without bias (except for the inclusion of PPI use in the GERD group). The purpose of defining control groups is to align with the specific objectives of different studies. A control group of healthy individuals is ideal for diagnostic studies, although a well-defined disease control group may also be very useful, depending on what we were studying.

Our study has strengths and limitations. The most important strengths are that we assessed strictly asymptomatic healthy individuals. Although it may not seem so at first glance, this is a significant advantage since all prior studies of the small intestine to date have used symptomatic controls, in whom the target disease being studied was ruled out. The reason for this selection bias, which was repeated until the present project, was the technical difficulty of collecting samples from asymptomatic healthy individuals. By describing the histology of the duodenal, gastric and oesophageal mucosa, as well as the morphometry and lymphocyte subpopulations in strictly asymptomatic healthy individuals, this study will enable diagnostic precision studies and establish normality cut-off points. Additionally, the inclusion of a GERD control group facilitates the validation of findings with greater ease.

The main limitation of our study is the presence of selection bias in the GERD patient group, as these patients present with acid reflux and higher PPI use, which prevents certain lymphocyte populations (CD45+CD3− cells, ILC, CD4+ and CD8+ in the lamina propria and NKT and ILC3 at the intraepithelial level) from being used as a “gold standard” since significant differences were observed between the GERD group and healthy controls. Despite this, we consider the GERD group to be a good control group in certain circumstances, as they showed normality in the remaining parameters and did not exhibit characteristic symptoms (e.g., diarrhoea, pain, or abdominal distension) of other diseases that may cause duodenal abnormalities (such as CD, Crohn’s disease, or IBS). Another limitation is the small sample size due to the technical challenges of including more individuals with very restrictive inclusion/exclusion criteria. However, a review of the literature reveals that studies aiming to characterise the duodenal mucosa of “healthy individuals” included approximately 20 participants. For instance, the classic studies by M. Hayat et al. [5], B. Veress et al. [35] and S. Pellegrino et al. [4] included 20, 18 and 14 “healthy controls”, respectively. Moreover, the majority of subjects in these studies were not asymptomatic, with most of them likely having functional bowel disorders, which was the primary reason for undergoing upper endoscopy. Nonetheless, the interquartile range of the different parameters assessed are narrow, indicating that the selected group is highly homogeneous. Thus, even if the sample size was increased, the results would likely not change.

## 4. Materials and Methods

### 4.1. Study Design, Definitions, Patients and Controls

This descriptive cross-sectional study was designed to characterise the normal mucosa of the upper digestive tract (oesophageal, gastric and duodenal) in healthy individuals. The inclusion criteria for the healthy control group were as follows: age >18 years, no comorbidities, provided written informed consent, negative symptom questionnaire (Appendix A), consuming a Mediterranean diet without restrictions, normal blood analysis (Appendix A), negative coeliac serology and Helicobacter Pylori tests, negative high-risk genetic testing for CD (only one allele of DQ2.2 or DQ7.5 was accepted separately), normal oesophagogastroduodenoscopy and histologically normal duodenal mucosa (<25 intraepithelial lymphocytes [IELs]) [3,4,5]. The exclusion criteria included age >65 years, body mass index >28, refusal to participate, severe disease (cardiopathy, lung and liver disease, coagulation disorders, neoplasms, etc.), personal or family history of CD and/or inflammatory bowel disease, pregnancy and/or breastfeeding, presence of any current digestive symptoms, potentially contagious diseases (human immunodeficiency virus, hepatitis C virus, hepatitis B virus, tuberculosis, COVID-19, etc.), travel to tropical countries in the previous 6 months, coagulopathy or anticoagulant treatment, use of any medication (including nonsteroidal anti-inflammatory drugs [NSAIDs]) in the previous 4 weeks, restrictive diets (vegetarian, vegan, or gluten-free), positive H. Pylori test, high-risk coeliac genetics (DQ2.5 or DQ8), positive coeliac serology, active smoking, alcohol consumption, abnormal oesophagogastroduodenoscopy or duodenal biopsy showing enteropathy.

Patients with GERD who met similar inclusion criteria were included as controls. However, in this case, proton pump inhibitor (PPI) use, the presence of H. pylori and active smoking were permitted, and upper gastrointestinal endoscopy was performed as part of routine clinical practice.

To account for possible discomfort, inconveniences and complications arising from upper gastrointestinal endoscopy with sedation, all healthy participants received compensation of €150. These subjects were blinded to which characteristics were required for inclusion in the study to avoid inducing negative responses in the questionnaire.

Additionally, to ensure participant safety, an insurance policy was secured from a leading provider in the field of clinical trial and biomedical research insurance for studies involving invasive procedures (Zurich Insurance Group Ltd., Zurich, Switzerland).

Only subjects who had negative responses on the dyspepsia questionnaire (Appendix A) were included.

Histopathological analysis, intestinal morphometry and flow cytometry were performed in a blinded manner. That means that the pathologists were blinded to the groups and their respective collected samples.

Considering sex and gender perspective, the data were disaggregated and analysed by sex in the results section [36].

### 4.2. Assessment of Subjects and Endoscopic Sample Collection

Before inclusion in the study, blood tests were performed to confirm the normality of the following parameters: complete blood count; general biochemistry, including renal and liver function and coagulation profile. Subsequently, an endoscopic procedure under sedation was performed with duodenal, gastric and oesophageal biopsies. Biopsy samples were obtained using 2.8 mm biopsy forceps (Radial Jaw 4, Boston Scientific, Marlborough, MA, USA).

For histopathological assessment of the duodenum, four endoscopic biopsies were collected from the second to third portions of the duodenum and two were taken from the duodenal bulb. From the gastric cavity, two endoscopic biopsies were collected from the antrum, and two biopsies were taken from the distal oesophagus.

For IEL assessment by flow cytometry, 14 biopsies were collected from the second portion of the duodenum.

### 4.3. Histopathological Assessment

In the upper digestive tract (oesophagus, stomach, and duodenum), the following cell types were evaluated: lymphocytes, eosinophils, and mast cells. Pathologists examined 5–10 high-resolution fields (40×) for each relevant section. To count intraepithelial lymphocytes, samples were processed using haematoxylin/eosin (H&E) staining, and the CD3 immunohistochemistry marker, prediluted anti-CD3 (2GV6) rabbit monoclonal antibody, was used (40×). We used the Marsh–Oberhuber classification [37] for the description of the duodenal mucosa.

Eosinophil counts were conducted using H&E staining (40×) and were performed in the duodenal villi (average eosinophil count with one decimal point in five well-oriented contiguous villi per high-power field [HPF]), in the duodenal and gastric lamina propria (number of eosinophils per HPF in well-oriented areas), and at the intraepithelial level in the oesophagus.

For mast cell counting, the immunohistochemistry marker prediluted anti-CD117 (EP10) (C-Kit) monoclonal antibody was used (40×), with counts performed in the duodenal villi (average with one decimal point in five well-oriented contiguous villi), in the duodenal and gastric lamina propria (number of mast cells per HPF in well-oriented areas), and at the intraepithelial level in the oesophagus.

In gastric samples, the presence of *H. pylori* was investigated using immunohistochemistry marker prediluted anti-*H. pylori* (SP48) rabbit monoclonal antibody.

The various immunohistochemical analyses mentioned were performed using the VENTANA platform (Roche Diagnostics, Basel, Switzerland).

### 4.4. Morphometric Assessment

For morphometric analysis, a high-resolution optical microscope at 40× magnification was used. Five well-oriented representative areas were analysed for the direct measurement of villus height and crypt depth, which serve as indicators of duodenal mucosal structure. This analysis was performed on biopsies from the second portion of the duodenum stained with H&E.

### 4.5. Coeliac Serology and HLA-DQ Genotyping

Serum IgA-tissue transglutaminase 2 antibodies (anti-tTG2) were analysed using a quantitative automated chemiluminescence kit (QUANTA FLASH h-tTG IgA kit; Inova Diagnostics, San Diego, CA, USA) with recombinant human TG2 from baculovirus as the antigen. Values > 20 CU were considered positive. Total serum IgA was measured by an immunoturbidimetry automated assay (Cobas 8000 c 207, Roche Diagnostics, Basel, Switzerland).

Genomic DNA from whole blood was purified using a commercial QIAamp DNA Blood Mini Kit (Qiagen, Düsseldorf, Germany). A commercial method of SSO-PCR to detect CD-associated HLA alleles (HLA-DQA1* and HLA-DQB1* alleles) was used (HISTO SPOT Coeliac Disease Kit, BAG Healthcare, Lich, Germany). DQ genotyping was performed according to previous recommendations [38].

### 4.6. Intestinal Lymphocyte Isolation and Quantification by Flow Cytometry

The samples for studying lymphocyte subpopulations were collected in a complete culture medium consisting of sterile advanced RPMI supplemented with 2% foetal bovine serum (FBS), 100% antibiotic–antimycotic mixture (10,000 U/mL penicillin, 10,000 µg/mL streptomycin, and 25 µg/mL amphotericin B) to prevent cell culture contamination, and 200 mM 1% L-glutamine for cell culture supplementation; both reagents were obtained from Gibco (Refs. 11570486 and 11500626; Thermo Fisher Scientific Inc., Waltham, MA, USA). Isolation of IELs was achieved by gentle rotation in an orbital shaker at 12 rpm for 90 min in a solution of 1 mM dithiothreitol and 1 mM ethylenediaminetetraacetic acid in 10% FBS Hanks balanced salt solution (HBSS) at room temperature [16]. The remaining tissue after IEL isolation was used for lamina propria lymphocyte (LPL) isolation by overnight incubation in the complete medium (walkout method). Total counts of IELs and LPLs were performed with trypan blue exclusion in a Neubauer chamber. A list of the antibodies used is available in Appendix A.

The following intraepithelial and LPL subpopulations were evaluated: TCRγδ+, CD3−, double-positive lymphocytes (DP, CD3+CD4+CD8+), double-negative lymphocytes (DN, CD3+CD4−CD8−), natural killer cells (NK, CD3−CD56+), natural killer T cells (NKT, CD3+CD56+), innate lymphoid cells (ILC1, ILC2, and ILC3), Vδ1+ T cells, Vδ2+ T cells, CD8α+CD8β−, and CD8α+CD8β+ cells.

The samples were acquired using an FACSCanto II or LSRFortessa flow cytometer (BD Bioscience, San Jose, CA, USA), and the data were analysed using BD FACSDiva v9.0 software (BD Biosciences, San Jose, CA, USA) and FlowJo v10.10 software (BD Biosciences, Ashland, OR, USA).

### 4.7. Statistical Analysis

Since this is an exploratory study, based on previous literature [4,5,35] and following a strict selection of potential healthy controls, we decided to include a minimum of 20 subjects of both sexes who fulfilled the inclusion criteria. Subsequently, we sought to include a similar number of cases from the GERD group, balanced by sex.

To compare characteristics between the healthy individuals and GERD groups, categorical variables are presented as the number of cases and percentages, whereas continuous variables are presented as medians and interquartile ranges (IQRs) or means and standard deviations (SDs). Density plots were used to compare the distributions of the subpopulations between healthy individuals and GERD patients. All analyses were conducted with a two-sided significance level of 0.05 using R software version 4.4.1 (https://www.r-project.org/).

## 5. Conclusions

In summary, this is the first study to describe the duodenal mucosa in healthy individuals and patients with GERD, establishing a “gold standard” of normality in the duodenal mucosa, which is fundamental for research on diseases such as CD, Crohn’s disease or IBS. The differences found between healthy controls and patients with GERD suggest that, whenever possible, healthy subjects should be included in research studies. Alternatively, we can include a well-defined homogenous control group, as it could include patients with GERD instead of a mixture of diseases.

## Figures and Tables

**Figure 1 ijms-26-01349-f001:**
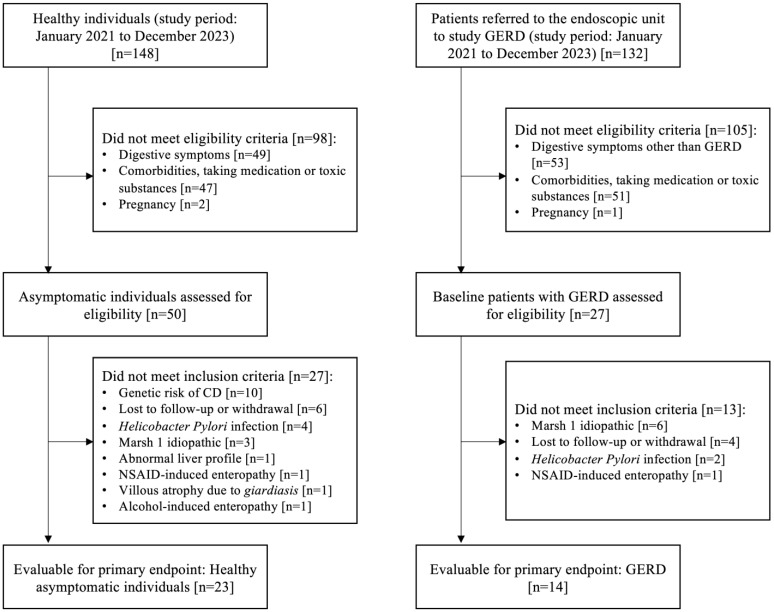
Study flow chart. Abbreviations: GERD: gastroesophageal reflux disease; NSAIDs: nonsteroidal anti-inflammatory drugs.

**Figure 2 ijms-26-01349-f002:**
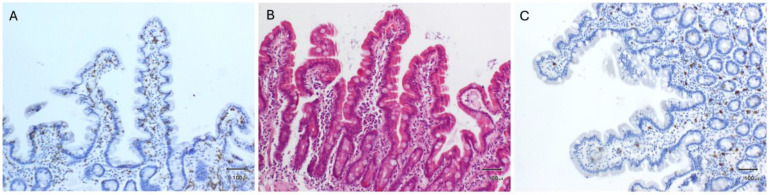
Histological images of duodenal samples from healthy individuals. (**A**) Immunohistochemical staining of CD3+ lymphocytes in the duodenal mucosa (Marsh 0). (**B**) H&E staining showing sparse eosinophils in the duodenal mucosa (Marsh 0). (**C**) Immunohistochemical staining of C-kit showing sparse mast cells in the duodenal mucosa (Marsh 0). Abbreviations: H&E: haematoxylin and eosin.

**Figure 3 ijms-26-01349-f003:**
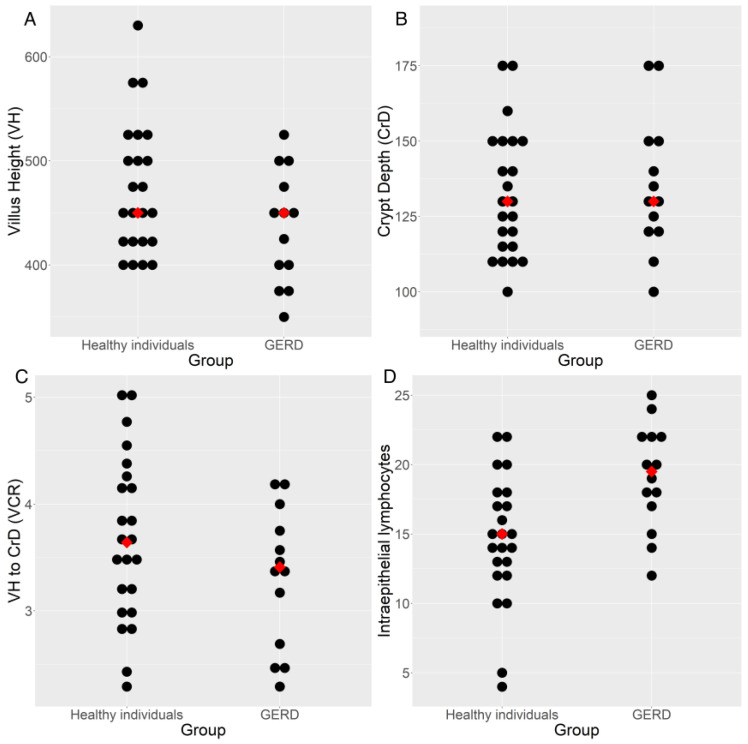
Strip chart of villus height (μm) (**A**), crypt depth (μm) (**B**), villus height-to-crypt depth ratio (VCR) (**C**) and intraepithelial lymphocytes (**D**) in the villi of healthy individuals compared with those in the gastroesophageal reflux disease (GERD) group. The red dot indicates the median.

**Figure 4 ijms-26-01349-f004:**
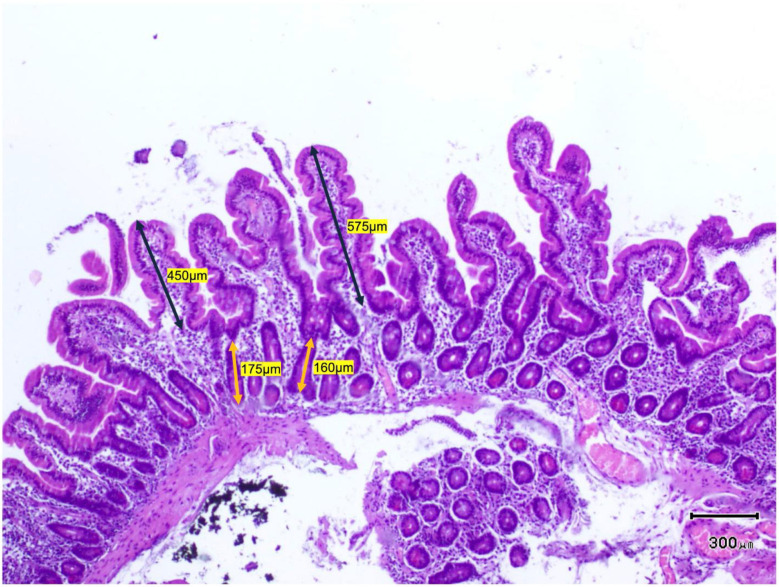
Haematoxylin and eosin staining of duodenal mucosa from healthy individuals (Marsh 0). The architecture of villi and crypts is preserved. The black arrows indicate two examples of villus height measurements (µm), and the orange arrows indicate two measurements of crypt depth (µm) to assess duodenal morphometry.

**Figure 5 ijms-26-01349-f005:**
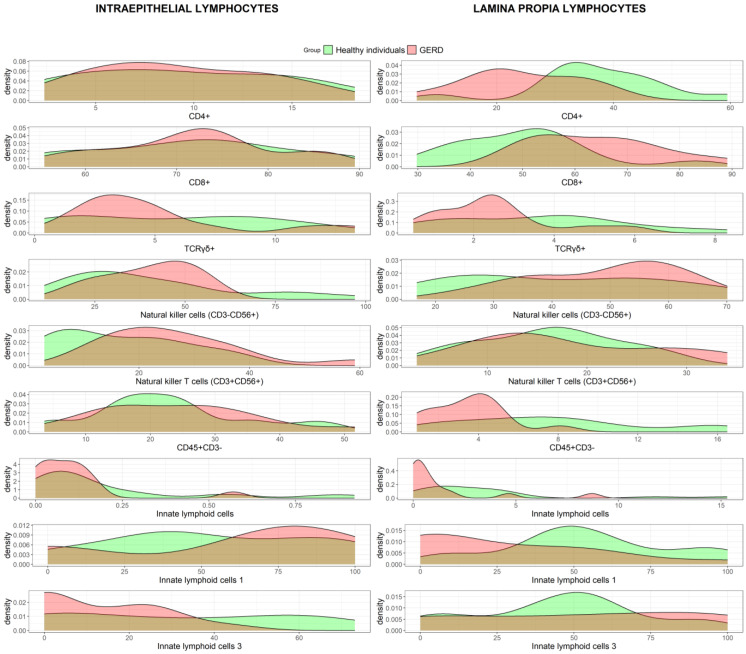
Main intestinal lymphocyte subpopulations in healthy individuals compared to those in the gastroesophageal reflux disease (GERD) group and the overlap between them.

**Figure 6 ijms-26-01349-f006:**
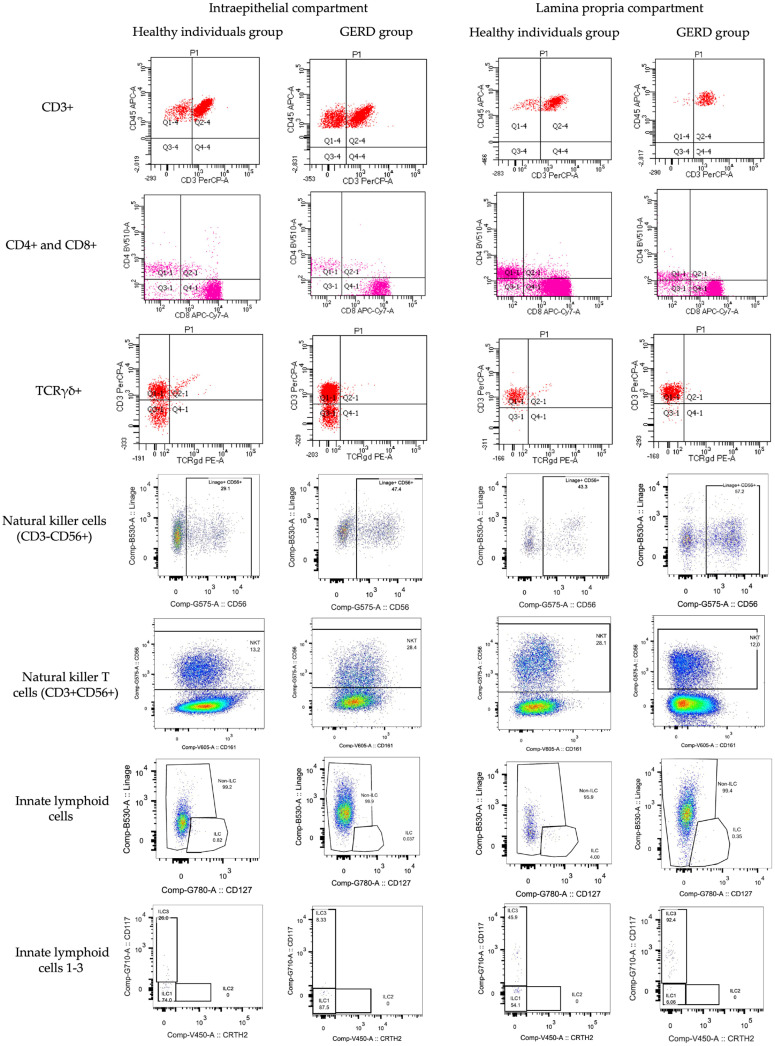
Intestinal cytometry panel of major intestinal lymphocyte subpopulations in healthy individuals compared to the gastroesophageal reflux disease (GERD) group.

**Table 1 ijms-26-01349-t001:** Baseline characteristics of the healthy control group and the gastroesophageal reflux disease (GERD) group.

Variables	Healthy Group (n = 23)	GERD Group (n = 14)
Age (years) ^a^	24.00 [21.00; 27.00]	31.00 [23.00; 37.00]
Female gender, n (%)	13 (56.52%)	8 (57.14%)
**Toxic Habits or Medication**		
Nonsmoker, n (%)	23 (100.00%)	12 (85.71%)
Former smoker, n (%)	0 (0.00%)	2 (14.29%)
Use of PPIs, n (%)	0 (0.00%)	11 (78.57%)
**HLA-DQ Genotype and Blood Count**		
HLA-DQ2.5, n (%)	0 (0.00%)	5 (50.00%)
HLA-DQ8, n (%)	0 (0.00%)	1 (10.00%)
HLA-DQ2.2, n (%)	5 (21.74%)	0 (0.00%)
HLA-DQ7.5, n (%)	0 (0.00%)	2 (20.00%)
HLA-DQ2 and HLA-DQ8-negative, n (%)	18 (78.26%)	2 (20.00%)
Haemoglobin ^b^	14.20 (1.36)	14.08 (1.51)
**Upper Gastrointestinal Endoscopy Findings**		
Normal, n (%)	19 (82.61%)	7 (50.00%)
Antritis, n (%)	2 (8.70%)	1 (7.14%)
Reflux oesophagitis, n (%)	0 (0.00%)	3 (21.43%)
Hiatal hernia, n (%)	0 (0.00%)	2 (14.30%)
Incompetent cardia, n (%)	0 (0.00%)	1 (7.14)
Gastric diverticulum, n (%)	1 (4.35%)	0 (0.00%)
Gastric polyp, n (%)	1 (4.35%)	0 (0.00%)

^a^ Median (interquartile range: 25%; 75%). ^b^ Mean +/− SD. Abbreviations: GERD: gastroesophageal reflux disease; PPIs: proton pump inhibitors; SD: standard deviation.

**Table 2 ijms-26-01349-t002:** Histological characteristics of the healthy and gastroesophageal reflux disease (GERD) groups.

Variables		Healthy Group (n = 23)	GERD Group (n = 14)	*p* Value ^b^
**Oesophageal Histology**				
Normal, n (%)		19 (82.6%)	11 (78.6%)	>0.999
Pathological, n (%)	Peptic oesophagitis, n (%)	1 (4.3%)	0 (0.0%)	0.200
	Idiopathic oesophagitis, n (%)	2 (8.7%)	0 (0.0%)	
	Eosinophilic oesophagitis, n (%)	0 (0.0%)	2 (14.3%)	
Oesophageal IEL count ^a^		16.50 [7.00; 37.00]	13.50 [6.00; 21.00]	0.464
Oesophageal EOS count ^a^		0.00 [0.00; 0.00]	0.00 [0.00; 0.00]	>0.999
Oesophageal MC count ^a^		1.00 [0.00; 3.00]	2.00 [1.00; 8.00]	0.187
**Gastric Histology**				
Normal, n (%)		19 (82.6%)	7 (50.0%)	0.063
Pathological, n (%)	Mild chronic gastritis, n (%)	4 (17.4%)	4 (28.6%)	0.236
	Gastritis due to *Helicobacter pylori*, n (%)	0 (0.0%)	3 (21.4%)	
Gastric IEL count ^a^		8.50 [6.00; 11.00]	9.00 [9.00; 13.00]	0.098
Gastric EOS count ^a^		4.50 [1.00; 8.00]	3.00 [1.00; 7.00]	0.566
Gastric MC count ^a^		30.00 [19.00; 39.00]	22.50 [12.00; 29.00]	0.132
**Duodenal Histology**				
Duodenal IEL count ^a^		15.00 [12.00; 18.00]	19.5 [17.00; 22.00]	0.005
Duodenal Intraepithelial EOS count ^a^		3.00 [2.00; 5.00]	3.00 [2.00; 4.00]	0.836
Duodenal lamina propria EOS count ^a^		14.00 [8.00; 28.00]	16.00 [12.00; 19.00]	0.863
Duodenal Intraepithelial MC count ^a^		3.80 [2.80; 5.60]	4.90 [3.20; 5.40]	0.424
Duodenal lamina propria MC count ^a^		30.00 [23.00; 35.00]	31.5 [25.00; 40.00]	0.415
Absence of duodenal parasites, n (%)		23 (100.00%)	14 (100.00%)	

^a^ Median (interquartile range: 25%; 75%). ^b^ Fisher’s exact test; Pearson’s chi-square test; Wilcoxon rank sum test. Abbreviations: IELs: intraepithelial lymphocytes; EOS: eosinophils; MCs: mast cells; GERD: gastroesophageal reflux disease.

**Table 3 ijms-26-01349-t003:** Histological characteristics of healthy individuals analysed by sex.

Variables		Male Sex (n = 10)	Female Sex (n = 13)	*p* Value ^b^
**Oesophageal Histology**				
Normal, n (%)		9 (90.0%)	10 (76.92%)	0.240
Pathological, n (%)	Peptic oesophagitis, n (%)	0 (0.0%)	1 (7.70%)	
	Idiopathic oesophagitis, n (%)	0 (0.0%)	2 (15.38%)	
Oesophageal IEL count ^a^		14.00 [9.00; 24.00]	21.00 [5.00; 42.00]	0.789
Oesophageal EOS count ^a^		0.00 [0.00; 0.00]	0.00 [0.00; 0.00]	0.486
Oesophageal MC count ^a^		1.00 [1.00; 1.00]	1.00 [0.00; 4.00]	>0.999
**Gastric Histology**				
Normal, n (%)		9 (90.00%)	10 (76.92%)	0.604
Pathological, n (%)	Mild chronic gastritis, n (%)	1 (10.00%)	3 (23.08%)	
Gastric IEL count ^a^		8.50 [7.00; 11.00]	8.00 [5.50; 12.00]	0.715
Gastric EOS count ^a^		3.00 [1.00; 7.00]	4.50 [2.50; 10.50]	0.207
Gastric MC count ^a^		25.00 [19.00; 32.00]	34.00 [25.00; 39.00]	0.321
**Duodenal Histology**				
Duodenal IEL count ^a^		15.00 [14.00; 18.00]	14.00 [12.00; 17.00]	0.686
Duodenal Intraepithelial EOS count ^a^		3.00 [1.00; 5.00]	4.00 [2.00; 5.00]	0.359
Duodenal lamina propria EOS count ^a^		12.50 [8.00; 23.00	15.00 [11.00; 28.00]	0.641
Duodenal Intraepithelial MC count ^a^		2.80 [2.60; 3.80]	4.80 [3.80; 6.00]	0.009
Duodenal lamina propria MC count ^a^		27.50 [18.00; 34.00]	30.00 [27.00; 35.00]	0.319

^a^ Median (interquartile range: 25%; 75%). ^b^ Fisher’s exact test; Wilcoxon rank sum test. Abbreviations: IELs: intraepithelial lymphocytes; EOS: eosinophils; MCs: mast cells.

**Table 4 ijms-26-01349-t004:** Crypt depth (μm), villus height (μm) and villus height-to-crypt depth ratio (VCR) in the villi of healthy individuals compared with those in the gastroesophageal reflux disease (GERD) group.

Variables	Healthy Group (n = 23)	GERD Group (n = 14)	*p* Value ^b^
Villus height (µm) ^a^	450.00 [425.00; 525.00]	450.00 [400.00; 475.00]	0.126
Crypt depth (µm) ^a^	130.00 [115.00; 150.00]	130.00 [120.00; 150.00]	0.691
Villus height to crypt depth ratio ^a^	3.64 [3.00; 4.26]	3.41 [2.69; 3.75]	0.193

^a^ Median (interquartile range: 25%; 75%). ^b^ Wilcoxon rank sum test. Abbreviations: IELs: Intraepithelial lymphocytes; GERD: gastroesophageal reflux disease.

**Table 5 ijms-26-01349-t005:** Intestinal lymphocyte subpopulations in healthy individuals compared with those in individuals with gastroesophageal reflux disease (GERD).

Variables/Groups	Intraepithelial Lymphocytes			Lamina Propria Lymphocytes		
	Healthy Individuals Group (n = 23) ^a^	GERD Group (n = 14) ^a^	*p* Value ^b^	Healthy Individuals Group (n = 23) ^a^	GERD Group (n = 14) ^a^	*p* Value ^b^
CD3+	73.80 [62.85; 77.20]	73.90 [64.50; 83.60]	0.509	91.40 [83.00; 93.70]	93.40 [88.10; 96.05]	0.136
CD4+ ^c^	9.11 [5.62; 13.70]	8.32 [5.17; 13.10]	0.951	35.65 [31.10; 42.40]	21.50 [19.40; 33.05]	0.008
CD8+ ^c^	71.85 [64.25; 78.50]	72.40 [64.50; 76.50]	>0.999	51.60 [39.80; 55.20]	64.80 [53.25; 71.00]	0.014
CD8α+CD8β− ^d^	39.40 [31.65; 53.90]	54.70 [39.70; 57.90]	0.157	43.65 [36.40; 69.00]	59.10 [46.30; 70.70]	0.274
CD8α+CD8β+ ^d^	60.60 [46.10; 68.35]	45.30 [42.10; 60.30]	0.157	56.35 [31.00; 63.60]	40.90 [29.30; 53.70]	0.274
Double Positive (CD4+CD8+) ^c^	8.76 [5.85; 11.30]	9.85 [7.47; 19.30]	0.087	10.40 [6.37; 12.40]	9.12 [6.32; 10.00]	0.354
Double Negative (CD4− CD8−) ^c^	9.26 [3.92; 13.95]	6.67 [1.99; 9.83]	0.123	2.42 [1.77; 3.40]	2.99 [0.69; 5.00]	0.857
TCRγδ+ ^c^	5.75 [1.70; 8.63]	3.95 [2.60; 5.40]	0.704	4.00 [1.70; 4.50]	2.40 [1.30; 2.60]	0.187
Vδ1+T cells ^e^	2.81 [1.02; 4.90]	1.80 [0.94; 8.45]	0.951	1.70 [0.55; 3.07]	1.32 [0.35; 3.67]	0.940
Vδ2+T cells ^e^	9.63 [3.67; 19.55]	7.42 [4.32; 12.10]	0.611	1.34 [0.98; 3.07]	1.98 [1.13; 3.08]	0.462
CD45+CD3− cells	21.97 [16.67; 26.40]	24.85 [15.20; 30.70]	0.834	7.80 [5.00; 9.60]	3.70 [2.30; 4.50]	0.007
Natural killer cells (CD3−CD56+) ^f^	32.20 [23.10; 52.00]	42.60 [31.00; 49.90]	0.471	40.00 [26.30; 52.50]	53.70 [38.20; 57.20]	0.129
Natural killer T cells (CD3+CD56+) ^c^	13.90 [6.00; 24.30]	24.00 [17.00; 34.50]	0.036	16.60 [9.94; 21.50]	14.95 [12.00; 27.10]	0.699
Innate lymphoid cells ^f^	0.11 [0.04; 0.25]	0.09 [0.03; 0.14]	0.308	2.40 [0.81; 4.00]	0.45 [0.09; 1.32]	0.007
Innate lymphoid cells 1 ^g^	50.00 [33.30; 85.70]	75.00 [0.00; 83.70]	0.705	53.70 [40.50; 68.80]	15.53 [0.00; 50.00]	0.012
Innate lymphoid cells 2 ^g^	0.00 [0.00; 0.00]	0.00 [0.00; 0.00]	0.461	0.00 [0.00; 0.00]	0.00 [0.00; 0.00]	
Innate lymphoid cells 3 ^g^	26.00 [0.00; 58.30]	7.15 [0.00; 25.00]	0.049	46.30 [31.20; 59.50]	53.55 [0.00; 92.40]	0.607

^a^ Median (interquartile range: 25%; 75%); ^b^ Wilcoxon rank sum test; Wilcoxon rank sum exact test; ^c^ of total CD3+; ^d^ of total CD8+; ^e^ of total TCRγδ+; ^f^ of total CD45+CD3− cells; ^g^ of total innate lymphoid cells. Abbreviations: CD: cluster of differentiation; GERD: gastroesophageal reflux disease.

## Data Availability

The database underlying this article is available upon reasonable request.

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
