# Peer review of "Histology of the Upper Gastrointestinal Tract, Morphometry and Lymphocyte Subpopulations of the Duodenal Mucosa: Insights from Healthy Individuals"

_ijms, 2025, doi:10.3390/ijms26031349_

Round 1

Reviewer 1 Report

Comments and Suggestions for Authors

1. The word “Histopathology” may be inappropriate, since the authors also analyzed healthy subjects. Maybe it should be replaced with "Histology"?

 2. In Conclusions, the authors say that clinical studies should include healthy controls. I don't know what their experience is, but healthy controls are included whenever possible. For a very long time

 3. The authors themselves say that one of the limitations of their study is the small number of respondents. And indeed, 23 healthy subjects and 14 with GERD represent a very small number to draw more reliable conclusions. Despite the high homogeneity of the investigated groups.  4. Overall, the author's idea is more than interesting, but I am of the opinion that more individuals should have been processed for more thorough conclusions. Unfortunately, this then suggests publishing these results much later.

Author Response

Reviewer #1

  1. The word “Histopathology” may be inappropriate, since the authors also analyzed healthy subjects. Maybe it should be replaced with "Histology"?

Response:

Thank you very much for this observation, you are absolutely right. We have changed “Histopathology” by “Histology” in the title and throughout the manuscript (page 1, lines 1 and 21). Thank you for pointing this out!

  1. In Conclusions, the authors say that clinical studies should include healthy controls. I don't know what their experience is, but healthy controls are included whenever possible. For a very long time

Response:
Thank you very much for this comment. We would like to expand on the explanation to clarify this point. Many studies evaluating the upper gastrointestinal tract refer to "healthy controls," but these are often not truly asymptomatic healthy individuals. Upon reviewing their methodology, it becomes evident that such studies frequently include patients who visited the hospital for some reason, typically mild digestive symptoms (e.g., abdominal distension, diarrhoea, mild abdominal pain). These individuals are ruled out for known organic diseases (e.g., cancer, inflammatory bowel disease, coeliac disease) and are often classified as having functional digestive disorders, such as irritable bowel syndrome (IBS).

Although some IBS have normal intestinal histology, many studies have demonstrated that many of these patients have some degree of low-grade mucosal inflammation, with mast cells playing a role in its pathophysiology. These patients have also frequently an increased number of intraepithelial lymphocytes of unknown aetiology (idiopathic lymphocytic enteritis). In summary, we consider that patients with irritable bowel syndrome are not a reliable gold standard for normality. For this reason, the aim of the present study was to recruit strictly healthy individuals to establish a true gold standard of normality in histology, morphometry, and the lymphocyte subpopulations studied.

In the present study, strictly asymptomatic healthy individuals were included. A special insurance policy was obtained to allow invasive procedures (such as upper gastrointestinal endoscopy) to be performed on these participants, as they had not sought medical care and were asymptomatic. Additionally, participants were compensated €150 for any inconvenience caused. Furthermore, to avoid selection bias, they were unaware that any detectable digestive symptoms or abnormal laboratory findings would disqualify them and make them ineligible for compensation.

Therefore, due to the difficulty of having healthy controls, we wanted to assess a group of patients with GERD, who at least theoretically could be better controls than patients with IBS. However, as demonstrated in our study, these patients also have subtle cellular abnormalities.

For all these reasons, we emphasise that this study indeed includes true healthy controls (asymptomatic, excluding diseases, toxins, drugs, etc). We have expanded the explanation in the discussion sections to clarify these points (page 14, lines 322-328).

  1. The authors themselves say that one of the limitations of their study is the small number of respondents. And indeed, 23 healthy subjects and 14 with GERD represent a very small number to draw more reliable conclusions. Despite the high homogeneity of the investigated groups.

  1. Overall, the author's idea is more than interesting, but I am of the opinion that more individuals should have been processed for more thorough conclusions. Unfortunately, this then suggests publishing these results much later.

Response:
Thank you very much for this comment. We will address points 3 and 4 together. As specified in the study limitations, we also acknowledge that the number of patients included is limited. However, we believe that our inclusion process was extremely rigorous (280 individuals were assessed, with 37 patients ultimately included over a two-year period). This strict selection allowed us to obtain a highly homogeneous group to be used as a gold standard for normality.

The main challenges in including more patients are both, technical (when assessing potential candidates properly, nearly everyone presents some type of digestive symptom, especially in older individuals) and economic (costs related to insurance, participant compensation and endoscopic procedures).

Despite these challenges, a review of the literature reveals very few similar studies involving "healthy individuals" undergoing invasive procedures to collect and evaluate samples. Moreover, the few available studies included approximately 20 participants. For instance, the classic studies by M. Hayat et al. (2002), B. Veress et al. (2004), and S. Pellegrino et al. (2011), which also aimed to establish the normality of the intestinal mucosa, included 20, 18, and 14 "healthy controls," respectively. Moreover, the majority of subjects included in these studies were not asymptomatic, with most of them probably having functional bowel disorders. That was the reason for what the upper endoscopy was performed.

For all these reasons, we consider that to have included more than 20 strictly asymptomatic healthy individuals is a significant achievement. We have added an explanation in the discussion section, citing the following articles to provide context regarding previously reported sample sizes, and we have also addressed this point in the study's limitations (page 14, lines 317-324).

Hayat, M.; Cairns, A.; Dixon, M.F.; O’Mahony, S. Quantitation of Intraepithelial Lymphocytes in Human Duodenum: What Is Normal? J Clin Pathol 2002, 55, 393–394, doi:10.1136/jcp.55.5.393.

Veress, B., Franzén, L., Bodin, L., & Borch, K. (2004). Duodenal Intraepithelial Lymphocyte-Count Revisited. Scandinavian Journal of Gastroenterology, 39(2), 138–144. https://doi.org/10.1080/00365520310007675

Pellegrino, S.; Villanacci, V.; Sansotta, N.; Scarfã, R.; Bassotti, G.; Vieni, G.; Princiotta, A.; Sferlazzas, C.; Magazzù, G.; Tuccari, G. Redefining the Intraepithelial Lymphocytes Threshold to Diagnose Gluten Sensitivity in Patients with Architecturally Normal Duodenal Histology. Aliment Pharmacol Ther 2011, 33, 697–706, doi:10.1111/j.1365-2036.2011.04578.x.

Reviewer 2 Report

Comments and Suggestions for Authors

This important paper aims to provide a reference standard for measurements related to the upper intestinal tract. It was possible to evaluate characteristics for 23 healthy subjects, which is an important result, since such data is sparse. In addition 14 GERD subjects were investigated to determine whether they can play the role of a reference standard for certain applications.

The amount of biological work is impressive, however I have a general comment related to the statistical analysis: in line 390 authors state "All analyses were conducted with a two-sided significance level of 0.05 using R software version 4.4.1". However, being not incorrect, it may be misleading, as most of the confidence intervals is provided by interquartile range, which is not that impressive and relates to confidence interval of only 0.5.

This relates to the conclusion about size of the sample, which authors have in line 266. In this line authors state "Nonetheless, the confidence intervals are narrow, indicating that the selected group is highly homogeneous." - however, the intervals need to be enlarged by factor of about 4 to represent at least CI=0.9.

Now the remaining of comments

a) the font in fig. 1 is not much readable (very small)

b) in line 112 - I believe that authors missed to state that differences in IEL are found in duodenum. Also the confidence intervals in the text and in the table don't match (probably due to rounding).

c) line 121 - "no differences were observed between the healthy and GERD groups in terms of villus morphometry (villus height [μm] and crypt depth [μm])" - but actually, the probability for no difference in height is only 0.126 (quite close to the treshold of 0.1), so maybe it would be good to point that this parameter could deserve some future investigation on a larger sample.

d) in figure 3 it looks like some distributions change from a "modal" distribution into equiprobable distribution. Maybe it would be a good idea to test samples also against distribution type?

e) in line 309 the word "normality" - suggests statistical meaning (like gaussian distribution), maybe it's not the best choice.

Author Response

Reviewer #2

This important paper aims to provide a reference standard for measurements related to the upper intestinal tract. It was possible to evaluate characteristics for 23 healthy subjects, which is an important result, since such data is sparse. In addition 14 GERD subjects were investigated to determine whether they can play the role of a reference standard for certain applications.

The amount of biological work is impressive, however I have a general comment related to the statistical analysis: in line 390 authors state "All analyses were conducted with a two-sided significance level of 0.05 using R software version 4.4.1". However, being not incorrect, it may be misleading, as most of the confidence intervals is provided by interquartile range, which is not that impressive and relates to confidence interval of only 0.5.

Response:
Thank you very much for these comments, as they have greatly contributed to improving the manuscript. We have reviewed the manuscript together with the co-authors statistics to correct the errors identified and clarify concepts to avoid misinterpretations.

Confidence intervals are not presented in this study. The interquartile range is reported using the first and third quartiles as a measure of variability for all continuous variables, except for haemoglobin (page 16, lines 445-449).

A two-sided significance level of 0.05 was used to determine statistical significance for Fisher’s exact test, Pearson’s chi-square test, and the Wilcoxon rank-sum test in the various analyses conducted. On page 17, line 450 of the statistics section, we have highlighted this aspect to address the concerns raised.

This relates to the conclusion about size of the sample, which authors have in line 266. In this line authors state "Nonetheless, the confidence intervals are narrow, indicating that the selected group is highly homogeneous." - however, the intervals need to be enlarged by factor of about 4 to represent at least CI=0.9.

Response:
Thank you very much for this comment; you are absolutely right. Accordingly, to what we explained in the previous point, the use of the words “confidence intervals” in the discussion of the paper is an error because we used interquartile range as a precision measure and not the confidence intervals. Thank you for identifying this mistake. To avoid confusion in interpretation, we have corrected the statement regarding sample size (page 14, lines 325): “Nonetheless, the interquartile range are narrow, indicating that the selected group is highly homogeneous. Thus, even if the sample size was increased, the results would likely not change.”

Now the remaining of comments

a) the font in fig. 1 is not much readable (very small)

Response:

We have increased the font size in Figure 1 to make it easier to read and interpret (page 3, lines 101-102).

b) in line 112 - I believe that authors missed to state that differences in IEL are found in duodenum. Also, the confidence intervals in the text and in the table don't match (probably due to rounding).

Response:
The differences in IEL count were found in the duodenum. We had forgotten in the text and was corrected accordingly to your observation (page 4, lines 112). Additionally, we have reviewed all confidence intervals and interquartile ranges to ensure consistency between the text and the tables (page 4, lines 112). Thank you for your valuable input.

c) line 121 - "no differences were observed between the healthy and GERD groups in terms of villus morphometry (villus height [μm] and crypt depth [μm])" - but actually, the probability for no difference in height is only 0.126 (quite close to the treshold of 0.1), so maybe it would be good to point that this parameter could deserve some future investigation on a larger sample.

Response:
From a statistical perspective, the medians are equal, and there is a high degree of overlap in the distributions (as shown in Figure 3 and Table 4, where—except for 3 points above and 3 points below—the remainder exhibits total overlap). However, it is true that the p-value is 0.1. We cannot determine whether an increase in sample size would render this difference significant, but the results are consistent with similar findings described in the literature by Rostami et al.

Furthermore, it aligns with the expectation that a condition like gastro-oesophageal reflux disease, which primarily affects the oesophagus, would not cause architectural lesions in the duodenal mucosa. As per your recommendation, we have added a comment in the discussion section specifying that with a larger sample size, it is highly improbable that differences emerged. In addition, the probability of a beta error is statistically stablished. However, it is advisable that studies assessing mucosal structure in the duodenum provide data on villous morphometry (page 12, lines 230-235).

Rostami, K.; Ensari, A.; Marsh, M.N.; Srivastava, A.; Villanacci, V.; Carroccio, A.; Asadzadeh Aghdaei, H.; Bai, J.C.; Bassotti, G.; Becheanu, G.; et al. Gluten Induces Subtle Histological Changes in Duodenal Mucosa of Patients with Non-Coeliac Gluten Sensitivity: A Multicentre Study. Nutrients 2022, 14, 2487, doi:10.3390/nu14122487.

d) in figure 3 it looks like some distributions change from a "modal" distribution into equiprobable distribution. Maybe it would be a good idea to test samples also against distribution type?

Response:
Thank you very much for this suggestion. Table 5 shows the comparison of subpopulations using the Wilcoxon rank-sum test (page 9-10, lines 157-159). To assess the distribution is not necessary, as subpopulations are compared using a non-parametric test that may be applied irrespective of the type of distribution.

e) in line 309 the word "normality" - suggests statistical meaning (like gaussian distribution), maybe it's not the best choice.

Response:

Thank you very much for this recommendation. We have revised the text, and the term "normality" has been replaced throughout with more precise terminology (e.g., "plausible range") to avoid confusion with the use of "normality" in reference to the distribution of a particular population.

Reviewer 3 Report

Comments and Suggestions for Authors

Thank you very much for the opportunity to review this important and needed  in the field manuscript. Indeed, having a good standard of normality is essential for research studies and a cornerstone for defining diagnostic criteria. I read the manuscript with great interest, however I have some comments and suggestions to augment it:

Results:

– Table 1. age is presented as mean (not median) - was age evenly distributed?

– since this is a histopathological study I would expect to see representative microscopic images as examples?!

– the morphological analysis did not include any sex- and age-dependant analysis? Why was it not considered? 

Discussion:

– among the GERD group, there were smokers, yet, this issue was not discussed?

Materials and Methods:

– Active smoking was among the exclusion criteria and at the same time there were active smokers in the GERD group (Table 1)?

– Why the Marsh-Oberhuber classification was decided to be used for the assessment of the duodenal mucosa?

– How many fields of vision were examined from each part from each patient (section 4.3 and 4.4)? 

– "Since this is an exploratory study, after a strict selection of potential healthy controls, we decided to include a minimum of 20 subjects of both sexes that fulfilled the inclusion criteria." - in the healthy group? please explain.

Author Response

Reviewer #3

Thank you very much for the opportunity to review this important and needed in the field manuscript. Indeed, having a good standard of normality is essential for research studies and a cornerstone for defining diagnostic criteria. I read the manuscript with great interest, however I have some comments and suggestions to augment it:

Results:

– Table 1. age is presented as mean (not median) - was age evenly distributed?

Response: You are right; in this case, age is better represented as the median (interquartile range: 25%; 75%) rather than the mean. We have corrected Table 1 accordingly: 24.00 [21.00; 27.00] for the healthy group and 31.00 [23.00; 37.00] for the gastro-oesophageal reflux disease group (page 3-4, lines 105-107).

– since this is a histopathological study I would expect to see representative microscopic images as examples?!

Response:

Thank you for this recommendation because this allows the readers to get familiar with the better stains to visualise different cell populations. We have decided to include three new figures:

  • Figure 2 (page 6): Displays histological images of duodenal samples from healthy individuals, including immunohistochemical staining of CD3+ lymphocytes, immunohistochemical staining of C-kit showing sparse mast cells, and haematoxylin and eosin (H&E) staining showing sparse eosinophils.
  • Figure 4 (page 8): Provides an example of duodenal morphometry, featuring H&E staining of duodenal villi from healthy individuals.
  • Figure 6 (page 11): Shows intestinal cytometry panels of major intestinal lymphocyte subpopulations in healthy individuals compared to the gastro-oesophageal reflux disease (GERD) group.

We believe these additions enhance the clarity and comprehensiveness of the manuscript.

– the morphological analysis did not include any sex- and age-dependant analysis? Why was it not considered? 

Response:

Thank you very much for your observation. Incorporating a sex and gender perspective in biomedical research is indeed very important. Following your recommendation, we have disaggregated and analysed the histological data by sex in the results section (Table 3, page 5). No significant differences were observed in the histological characteristics studied between men and women in healthy individuals, except for the duodenal mast cell count that were increased in women (p=0.009) (page 4, lines 119-124). This finding is consistent with previous literature: studies such as those by Barbara et al. (2004) and Cremon et al. (2011) that also identified a significant increase in the number of mast cells in the colonic mucosa, predominantly in women with irritable bowel syndrome (IBS). Physiologically, mast cells express receptors for oestrogen and progesterone, suggesting that these hormonal mediators could modulate their activity. This phenomenon may be related to hormonal fluctuations during the menstrual cycle in women, contributing to a higher mast cell density in the female intestinal mucosa. Additionally, no significant differences were found in histological characteristics when the data were analysed by age groups (age groups defined by the median ± 25 years) in healthy individuals (page 12, lines 219-229). References:

Barbara, G., Stanghellini, V., de Giorgio, R., Cremon, C., Cottrell, G. S., Santini, D., Pasquinelli, G., Morselli-Labate, A. M., Grady, E. F., Bunnett, N. W., Collins, S. M., & Corinaldesi, R. (2004). Activated Mast Cells in Proximity to Colonic Nerves Correlate with Abdominal Pain in Irritable Bowel Syndrome. Gastroenterology, 126(3), 693–702. https://doi.org/10.1053/j.gastro.2003.11.055

Cremon, C., Carini, G., Wang, B., Vasina, V., Cogliandro, R. F., de Giorgio, R., Stanghellini, V., Grundy, D., Tonini, M., de Ponti, F., Corinaldesi, R., & Barbara, G. (2011). Intestinal serotonin release, sensory neuron activation, and abdominal pain in irritable bowel syndrome. American Journal of Gastroenterology, 106(7), 1290–1298. https://doi.org/10.1038/ajg.2011.86

Additionally, we have reviewed and included the following citation to address this perspective:

Heidari, S.; Babor, T.F.; De Castro, P.; Tort, S.; Curno, M. Sex and Gender Equity in Research: Rationale for the SAGER Guidelines and Recommended Use. Res Integr Peer Rev 2016, 1, doi:10.1186/s41073-016-0007-6.

Discussion:

– among the GERD group, there were smokers, yet, this issue was not discussed?

Materials and Methods:

– Active smoking was among the exclusion criteria and at the same time there were active smokers in the GERD group (Table 1)?

Response:
Thank you very much for this observation; it has been very helpful in improving the manuscript. We will address these two points jointly.

The exclusion criterion for smokers was applied only to the group of healthy individuals and not to the gastro-oesophageal reflux disease (GERD) group. Following your observation, we have corrected this error and clearly specified it in the methodology to avoid confusion (page 15, line 350-352). Only two social smokers were included in the GERD group, each consuming <5 cigarettes per week, whereas there were no smokers in the healthy individuals group.

The most permissive inclusion criteria with the GERD group than for healthy controls emphasise the difficulty of finding true “healthy” controls among patients with several gastrointestinal complaints and diseases.

– Why the Marsh-Oberhuber classification was decided to be used for the assessment of the duodenal mucosa?

Response:
That is an excellent question—what is the best classification to categorise pathological states of the duodenal mucosa? This issue has been widely discussed in the scientific literature, and we have included two references in the manuscript to clarify this topic.

We decided to use the Marsh-Oberhuber classification because it is one of the most well-known and widely cited classifications worldwide. But there are, other classifications, such as that proposed by Ensari et al. in 2010, that are more simple and therefore probably more reproducible.

Peña, A.S. What Is the Best Histopathological Classification for Celiac Disease? Does It Matter? Gastroenterol Hepatol Bed Bench 2015, 8, 239–243.

Walker, M.M.; Murray, J.A. An Update in the Diagnosis of Coeliac Disease. Histopathology 2010, 59, 166–179, doi:10.1111/j.1365-2559.2010.03680.x.

– How many fields of vision were examined from each part from each patient (section 4.3 and 4.4)? 

Response:
Thank you very much for this comment. In this study, pathologists examined 5–10 high-resolution fields (40x) for each relevant section of the oesophageal, gastric, and duodenal samples (page 15, line 383). For intraepithelial lymphocytes, counts were performed on a representative stretch of 100 enterocytes in each field. For duodenal morphometry, five well-oriented representative areas were analysed for structural assessment. These specifications have been added to the methodology section (page 16, line 406).

– "Since this is an exploratory study, after a strict selection of potential healthy controls, we decided to include a minimum of 20 subjects of both sexes that fulfilled the inclusion criteria." - in the healthy group? please explain.

Response:
The aim of the present study was to evaluate the upper digestive tract of strictly healthy individuals to establish a gold standard. While drafting this project, we conducted an extensive literature review and found that most studies included symptomatic patients (predominantly those with irritable bowel syndrome) as "healthy controls".

The few available studies included approximately 20 participants. For instance, the classic studies by M. Hayat et al. (2002), B. Veress et al. (2004), and S. Pellegrino et al. (2011), which also aimed to establish the normality of the intestinal mucosa, included 20, 18, and 14 "healthy controls," respectively. However, these individuals were generally not strictly asymptomatic.

For all these reasons, we consider it a significant achievement to have included more than 20 strictly asymptomatic healthy individuals in our study.

Additionally, we were extremely rigorous in our inclusion criteria: 280 individuals were assessed, with 37 patients ultimately included over a two-year period. This strict selection allowed us to obtain a highly homogeneous group to be used as a gold standard for normality.

Subsequently, we sought to include a similar number of cases from the gastro-oesophageal reflux disease (GERD) group, balanced by sex. However, we were unable to balance the groups by age, as the GERD group was older (p=0.022). This is because it was challenging to recruit younger patients with GERD, as younger individuals are typically healthier and less likely to have such symptoms. Conversely, it was also difficult to include strictly healthy older individuals, which is consistent with the observation that older individuals are more likely to present symptoms or comorbidities that exclude them from being classified as strictly healthy.

We have added an explanation in the methods (page 16, line 444-448) and discussion section (page 14, lines 317-325), citing the following articles to provide context regarding previously reported sample sizes, and we have also addressed this point in the study's limitations.

Hayat, M.; Cairns, A.; Dixon, M.F.; O’Mahony, S. Quantitation of Intraepithelial Lymphocytes in Human Duodenum: What Is Normal? J Clin Pathol 2002, 55, 393–394, doi:10.1136/jcp.55.5.393.

Veress, B., Franzén, L., Bodin, L., & Borch, K. (2004). Duodenal Intraepithelial Lymphocyte-Count Revisited. Scandinavian Journal of Gastroenterology, 39(2), 138–144. https://doi.org/10.1080/00365520310007675

Pellegrino, S.; Villanacci, V.; Sansotta, N.; Scarfã, R.; Bassotti, G.; Vieni, G.; Princiotta, A.; Sferlazzas, C.; Magazzù, G.; Tuccari, G. Redefining the Intraepithelial Lymphocytes Threshold to Diagnose Gluten Sensitivity in Patients with Architecturally Normal Duodenal Histology. Aliment Pharmacol Ther 2011, 33, 697–706, doi:10.1111/j.1365-2036.2011.04578.x.

Round 2

Reviewer 3 Report

Comments and Suggestions for Authors

Good work! I have no other comments!